# Conferring High IAA Productivity on Low-IAA-Producing Organisms with PonAAS2, an Aromatic Aldehyde Synthase of a Galling Sawfly, and Identification of Its Inhibitor

**DOI:** 10.3390/insects14070598

**Published:** 2023-07-02

**Authors:** Takeshi Hiura, Hibiki Yoshida, Umi Miyata, Tadao Asami, Yoshihito Suzuki

**Affiliations:** 1United Graduate School of Agricultural Science, Tokyo University of Agriculture and Technology, 3-5-8 Saiwai-cho, Fuchu-shi, Tokyo 183-0054, Japan; s215976w@st.go.tuat.ac.jp; 2Department of Food and Life Sciences, College of Agriculture, Ibaraki University, 3-21-1 Chuo, Ami-machi, Inashiki-gun, Ibaraki 300-0393, Japan; 3Department of Applied Biological Chemistry, Graduate School of Agricultural and Life Sciences, The University of Tokyo, 1-1-1 Yayoi, Bunkyo-ku, Tokyo 113-8657, Japan; asami@g.ecc.u-tokyo.ac.jp

**Keywords:** phytohormone, auxin, insect gall, aromatic aldehyde synthase, inhibitor

## Abstract

**Simple Summary:**

The gall-inducing sawfly (*Pontania* sp.) possesses high concentrations of indoleacetic acid (IAA; the active form of the phytohormone auxin), which may play an important role in gall induction. Among the insect aromatic aldehyde synthases (AASs) studied to date, sawfly PonAAS2 is the only AAS involved in IAA biosynthesis that produces indoleacetaldehyde from Trp. In this study, we show that the introduction of the *PonAAS2* gene is able to confer high IAA productivity on other organisms, using *Caenorhabditis elegans* as the model system. We also identified a specific inhibitor of PonAAS2.

**Abstract:**

Gall-inducing insects often contain high concentrations of phytohormones, such as auxin and cytokinin, which are suggested to be involved in gall induction, but no conclusive evidence has yet been obtained. There are two possible approaches to investigating the importance of phytohormones in gall induction: demonstrating either that high phytohormone productivity can induce gall-inducing ability in non-gall-inducing insects or that the gall-inducing ability is inhibited when phytohormone productivity in galling insects is suppressed. In this study, we show that the overexpression of *PonAAS2*, which encodes an aromatic aldehyde synthase (AAS) responsible for the rate-limiting step in indoleacetic acid (IAA) biosynthesis in a galling sawfly (*Pontania* sp.) that contains high levels of endogenous IAA, conferred high IAA productivity on *Caenorhabditis elegans*, as the model system. This result strongly suggests that PonAAS2 can also confer high IAA productivity on low-IAA-producing insects. We also successfully identified an inhibitor of PonAAS2 in a chemical library. This highly selective inhibitor showed stronger inhibitory activity against AAS than against aromatic amino acid decarboxylase, which belongs to the same superfamily as AAS. We also confirm that this inhibitor clearly inhibited IAA productivity in the high-IAA-producing *C. elegans* engineered here.

## 1. Introduction

Insects of various taxonomic groups can induce galls in host plants. Gall morphologies and the induction processes involved are complex and diverse, and the mechanisms underlying these processes have yet to be clarified. However, because many gall-inducing insects express higher concentrations of phytohormones, such as indole-3-acetic acid (IAA), an active form of auxin, and cytokinins than the host plant tissues, the involvement of these phytohormones in gall induction has been proposed [1]. Sawfly (*Pontania* sp.) and other insects have been shown to convert l-tryptophan (Trp) to IAA [2,3,4]. However, non-gall-inducing insects, such as the silkworm *Bombyx mori*, also contain endogenous IAA, and crude enzyme solutions prepared from *B. mori* also contain activity that converts Trp to IAA [5]. Therefore, we hypothesized that insects originally had the ability to synthesize IAA, and that the sawfly and other insects exploited this ability for gall induction by enhancing their capacity for IAA production. Based on this hypothesis, we aimed to clarify the biosynthetic mechanism involved using *B. mori* and the galling sawfly. We identified indole-3-acetaldoxime (IAOx) and indole-3-acetaldehyde (IAAld) as the biosynthetic intermediate compounds that were converted to IAA by the crude enzymes of both *B. mori* and the sawfly. Furthermore, we identified an inhibitor from a chemical library that inhibits the conversion of IAAld to IAA [5]. Using this inhibitor, we showed that the conversion of IAAId to IAA is an intermediate step in the conversions of both Trp and IAOx to IAA. Based on the apparent conversion activity of the crude enzyme preparation for each candidate precursor, we tentatively assumed the biosynthetic pathway from Trp to IAA to be Trp→IAOx→IAAld→IAA or Trp→IAAld→IAA [6]. We then purified and identified an aldehyde oxidase from *B. mori* as the enzyme responsible for the conversion of IAAld to IAA [7]. In the course of that study, we found that IAAld is non-enzymatically formed from Trp in the presence of flavin adenine dinucleotide (FAD) [8], a coenzyme of many oxidoreductases, including aldehyde oxidase, suggesting that non-enzymatic reactions may also be involved in IAA synthesis via the crude enzyme solution from *B. mori* in vitro [7].

However, because the galling sawfly converts Trp to IAAld more efficiently than *B. mori* [5], we searched for the enzymes responsible for the rate-limiting steps upstream from IAAld in the sawfly, based on its genetic information. We identified PonFMO1 as the enzyme responsible for the Trp→IAOx conversion and PonAAS2 as that for Trp→IAAld [9]. However, PonFMO1 was very weakly active, suggesting that PonAAS2 plays a major role in IAA production. PonAAS2 belongs to the aromatic aldehyde synthase (AAS) enzyme family, which catalyzes aromatic amino acids to their corresponding aldehydes in a bifunctional catalysis, which involves decarboxylation followed by oxidative deamination. Among the AASs in insects, those of *Drosophila melanogaster*, *B. mori*, *Aedes aegypti*, and certain other mosquito species have been characterized, and their major substrate has been shown to be l-DOPA [10,11]. These AASs have no Trp-converting activity and, to date, PonAAS2 is the only known insect AAS that uses Trp as a substrate. Therefore, the acquisition of PonAAS2 may have led to the high IAA productivity in the galling sawfly.

It has been reported that many terrestrial arthropods, including spiders and mites, possess IAA but that there is no significant correlation between endogenous IAA levels and gall-inducing ability [12]. This fact does not immediately negate the role of IAA in gall induction. However, no study has provided solid evidence for the importance of IAA in gall induction. The involvement of IAA and cytokinins in sawfly-induced galls is supported not only by the high concentrations of those phytohormones in the insect body but also by an expression analysis of phytohormone activity marker genes in the host plant. The callus-like tissue inside the gall expresses higher levels of genes encoding catabolizing enzymes that inactivate the phytohormones than the leaf tissue of the host willow plant (*Salix japonica* Thunb.) and higher levels of genes encoding negative regulators of phytohormone signal transduction pathways [2]. However, even in this case, there is no direct evidence that phytohormones are involved in gall induction. To confirm the roles of phytohormones in gall induction experimentally, two approaches can be considered. In one, non-gall-inducing insects can be provided with a strong capacity to synthesize phytohormones; in the other, the ability of gall-inducing insects to produce phytohormones is disrupted. In this study, we utilized the *PonAAS2* gene to provide high IAA productivity at the level of individual organisms in a model animal system and investigated PonAAS2 inhibitors.

As shown in this study, *Caenorhabditis elegans* has negligible levels of endogenous IAA, and its conversion activity for Trp→IAA is very low, whereas like many insects [9], it shows strong IAAld→IAA conversion activity. Therefore, *C. elegans* is an ideal model in which to investigate whether the introduction of *PonAAS2* can confer high IAA productivity on individual organisms.

We reported a PonAAS2-inhibitor in a previous study, but it showed weak inhibitory activity only at high concentrations [9]. Therefore, we were concerned that using the inhibitor at high concentrations would adversely affect the general physiology of inhibitor-treated insects. Moreover, because AAS shares very high homology with aromatic amino acid decarboxylase (AAAD) [13], which is essential for the production of certain key neurotransmitters [14], eggshell tanning [15], cuticle hardening [16,17], and immune responses [18,19,20], the specificity of any inhibitor of AAS is critical. Therefore, we searched again for inhibitors with strong inhibitory activity and strong selectivity for AAS of the galling sawfly.

## 2. Materials and Methods

### 2.1. Chemicals

Labeled tryptophan ([^13^C_11_, ^15^N_2_]l-Trp) was obtained from Sigma Aldrich (St. Louis, MO, USA). l-Tryptophan (Trp), 3-(3,4-dihydroxyphenyl)-l-alanine (DOPA), l-phenylalanine (Phe), l-tyrosine (Try), tryptamine, dopamine, 2-phenylethylamine, tyramine, and 5-hydroxytryptamine (serotonin) were purchased from Fujifilm Wako Chemicals (Osaka, Japan); 5-hydroxy-l-tryptophan (5-HTrp) and indole-3-acetaldehyde-sodium bisulfite derivative from Sigma-Aldrich; and dihydroxyphenylacetaldehyde (DHPAAld) was purchased from Cayman Chemical (Ann Arbor, MI, USA). Carbidopa and α-methyldopa (AMD) were purchased from TCI chemicals (Tokyo, Japan) and α-methyl-dl-tryptophan from Merck (Burlington, MA, USA). The Maybridge chemical library (Thermo Fisher Scientific, Waltham, MA, USA) was screened for inhibitors of PonAAS2. 2-[2-(3,4-Dihydroxyphenyl)-2-oxoethyl]-1H-1,2-benzisothiazole-1,1,3(2H)-trione (HTS09643) and 2-[3-(2,3-dihydro-1-benzofuran-5-yl)-1H-1,2,4-triazol-5-yl]phenol (HTS10550) were also obtained from Thermo Fisher Scientific.

### 2.2. Construction of a Plasmid Vector for the Expression of PonAAS2::GFP in C. elegans

The PonAAS2::GFP expression vector was constructed in pUC118. The region encoding green fluorescent protein (GFP) was amplified by PCR from pPV477 (Addgene #42930, a gift from James Lok) and cloned into the *Bam*HI and *Sal*I sites of pUC118 (pUC–GFP). The *unc-54* terminator was amplified from pSA108 (Addgene #59786, a gift from Jeremy Nance [21]) and cloned into the *Mlu*I and *Sal*I sites of pUC–GFP (*Sal*I site was integrated into the reverse primer for GFP amplification) (pUC–GFP–ter). The *cdc-42* promoter region was amplified from pSA108 and cloned into the *Kpn*I and *Bam*HI sites of pUC–GFP–ter, upstream from the GFP coding region (pUC–pro–GFP–ter). The PonAAS2 insert with *Bln*I and blunt-ended *Apa*I sites prepared from a plasmid containing the *PonAAS2* gene [9] was cloned into the *Bln*I and blunt-ended *Bam*HI sites of pUC–pro–GFP–ter on the 5’ side of GFP (pUC–pro–PonAAS2::GFP–ter). The primers used for the above PCRs are shown in Appendix A. The expression plasmid for mCherry under the control of the *myo-2* promoter (pCFJ90) was obtained from Addgene (#19327, a gift from Erik Jorgensen [22]).

### 2.3. Expression of PonAAS2::GFP in C. elegans

*Caenorhabditis elegans* strain N2 and *Escherichia coli* strain OP50-1 (stm^r^) were obtained from the National BioResource Project (NBRP). *Caenorhabditis elegans* worms were subcultured at 20 °C on NGM agar medium, on which *E. coli* OP50-1 had been grown. The plasmids were injected into the gonads of young adult worms, and the worms were recovered after injection using well-established methods [23,24]. The concentration of pCFJ90 was adjusted to 2.5 ng/μL and that of pUC–pro–PonAAS2::GFP–ter to 200, 500, or 1000 ng/μL. The microinjection needles were prepared from borosilicate glass capillaries (GD-1, Narishige, Tokyo, Japan) using a capillary pulling device (PN-30, Narishige). The needle tip was opened with a microforge (MF-900, Narishige), and the opened tip was sharpened using a grinder (EG-400, Narishige). After the plasmid vector was injected, the worms were restored in M9 buffer. A stereomicroscope (SZX-10, Olympus, Tokyo, Japan) equipped with a coaxial epifluorescence system (SZX-RFL2, Olympus) was used to select lines of *C. elegans* showing fluorescence. F1 individuals showing both mCherry red fluorescence and GFP green fluorescence were selected, and in subsequent generations, those with GFP fluorescence were selected and cultured successively. Photographs of the fluorescent worms were taken using a fluorescence microscope (APEXVIEW APX100, Olympus).

### 2.4. Protein Analysis of C. elegans Expressing PonAAS2::GFP

Wild-type worms and those expressing PonAAS2::GFP were collected in 2 mL of M9 buffer from three 6 cm NGM plates containing about 100 adults for each line. The worm suspension was transferred into a 15 mL tube and the worms were pelleted via centrifugation at 100× *g* for 2 min. The worms were washed twice with 2 mL of M9 buffer by centrifugation and the supernatant was removed. The worms were gently suspended again in 30% (*w*/*v*) sucrose and centrifuged at 100× *g* for 2 min. The floating worms were collected in a new 15 mL centrifuge tube with a micropipette fitted with a cut-off tip. After suspension in M9 buffer and centrifugation, the worms were collected as a pellet. Most of the supernatant was removed, and the worms were resuspended with a small amount of the remaining buffer (~200 μL) and transferred to a 1.5 mL tube. After centrifugation at 200× *g* for 2 min, the total volume was adjusted to about 50 μL. The worm suspension was mixed with an equal volume of 2× sample buffer for SDS-PAGE and incubated at 95 °C for 7 min. After centrifugation at 20,630× *g* for 15 min, the supernatant was recovered as the soluble protein fraction. An aliquot (10 μL) of this fraction was separated with SDS-PAGE on 10% acrylamide gel using a conventional method. The acrylamide gel was either stained with Coomassie Brilliant Blue or subjected to Western blot analysis. For the Western blot analysis, the proteins separated with SDS-PAGE were transferred to a nitrocellulose membrane (Hybond ECL, GE Healthcare Life Sciences, Boston, MA, USA). The membrane was sequentially incubated with a 1000-fold-diluted anti-GFP monoclonal antibody (mFX75, Fujifilm Wako Chemicals, Osaka, Japan), 5000-fold-diluted rabbit anti-mouse immunoglobulin G antibody conjugated with horseradish peroxidase (Fujifilm Wako Chemicals), and luminescent reagent (ECL™ Prime Western Blotting Detection System, GE Healthcare Life Sciences). Luminescence was detected using a chemiluminescence imaging platform (Fusion SL4, Vilber Loumart, Marne-la-Vallée, France).

### 2.5. Large-Scale Analysis of Endogenous IAA and Conversion Activity in Wild-Type C. elegans

About ten young adult worms were transferred to a 9 cm NGM plate and incubated for 1 week at 20 °C. A mixture of different-stage larvae and adults was recovered from three plates, washed with M9 buffer as described above in Section 2.4, and combined. The final volume was adjusted to ~2.5 mL using M9 buffer, and an aliquot (150 μL) was used for the conversion experiments or the analysis of endogenous IAA. For the conversion experiments, either 10 μg of [^13^C_11_,^15^N_2_]Trp or 800 ng of IAAld was added to the worm suspension and incubated for 17 h at 20 °C in the dark. Worm suspensions treated at 95 °C for 5 min were also used to check for non-enzymatic conversion. To investigate whether the *E. coli* in the worm bodies that the worms fed on were a source of IAA or involved in the production of IAA from Trp or IAAld, 150 μL of an *E. coli* suspension at an optical density at a wavelength of 600 nm (OD_600_) of 0.5 was also analyzed for endogenous amounts of IAA and used for the conversion experiments. Each sample was adjusted to 75% methanol and 5% formic acid and spiked with 1 ng of [^13^C_6_]IAA as the internal standard. After homogenization with a bead homogenizer at 2800 rpm for 1 min (μT-12, Taitec, Saitama, Japan) and centrifugation at 20,630× *g* for 5 min, the supernatant was passed through an Oasis HLB column (1 mL/30 mg; Waters) equilibrated with 75% methanol and 5% formic acid and eluted with 1 mL of the same solution. All the eluates were combined, concentrated, and subjected to a liquid chromatography–tandem mass spectrometry (LC–MS/MS) analysis, as described previously [6]. All the experiments were performed in triplicate, and the data are shown as means ± standard deviations (SD).

### 2.6. Analysis of Endogenous IAA and Conversion Activity in C. elegans Expressing PonAAS2::GFP

Three adults of each line showing green fluorescence caused by the expression of PonAAS2::GFP and of the wild type were transferred to 1.5 mL tubes containing 10 μL of M9 buffer. For the conversion experiment, the worms were homogenized via pipetting, and 500 ng of [^13^C_11_,^15^N_2_]Trp was added to the tube, which was then incubated for 12 h. For the analysis of endogenous IAA or [^13^C_10_,^15^N_1_]IAA converted from [^13^C_11_,^15^N_2_]Trp, 90 μL of methanol containing 1 ng of [^13^C_6_]IAA was added and sonicated for 30 s. After centrifugation at 20,630× *g* for 5 min, the supernatant was recovered, concentrated, and subjected to LC–MS/MS analysis, as described previously [6]. All the experiments were performed in triplicate, and the data are shown as means ± SD.

### 2.7. Screening a Chemical Library for Inhibitors of PonAAS2

Recombinant PonAAS1 and PonAAS2 were prepared as fusion proteins with trigger factor, using pCOLD TF DNA (Takara Bio), as described previously [9]. The initial screening of a chemical library for PonAAS2 inhibitors was as described previously, using 2 μg of each compound in 50 μL of reaction mixture [9], because the stock solutions of all the compounds had not been prepared as equal moles/volume but as equal weights/volume. In the second round of screening, the inhibitory activities of the candidate compounds were confirmed at various molar concentrations. The concentration of all substrates was 1 mM in the reactions except for the kinetic analysis.

## 3. Results

### 3.1. PonAAS2 Confers High IAA Productivity on C. elegans

Because the transgenesis system is well established in *C. elegans*, we examined whether this organism could be used to demonstrate that the *PonAAS2* gene increases IAA productivity in individual organisms. First, the endogenous level of IAA was found to be almost negligible in *C. elegans*, even when plenty of worms (equivalent to about 0.2 plates saturated with worms) were assayed. Moreover, whereas the Trp→IAA conversion activity was very low, IAAld was efficiently converted to IAA enzymatically (Figure 1). *Escherichia coli*, used as food for the worms, contained no endogenous IAA and showed no conversion activity. These results indicate that if PonAAS2 functions in *C. elegans*, it can potentially confer IAA productivity on the organism.

We constructed a vector from which PonAAS2 was expressed as a fusion protein with GFP under the control of the relatively constitutive *cdc-42* (cell division cycle 42) promoter. An mCherry expression vector was used to select the specimens in which the expression vector had been successfully injected into the gonads. The mixture of these vectors was injected into the gonads of young adult *C. elegans* (F0 generation). The concentration of the mCherry expression vector was fixed at 2.5 ng/μL for its transient expression in the F1 larvae, and the PonAAS2::GFP expression vector was used at several concentrations. Figure 2a shows F1 progeny displaying the red fluorescence of mCherry. When the PonAAS2 expression vector was used at a concentration of 200 ng/μL, a few of these lines continued to show the green fluorescence of GFP for more than several generations. Three such lines were established. Because no fluorescence was observed in some progeny of these lines during passage, we inferred that the expression cassette was not integrated into the genome but was maintained as an extrachromosomal array [23]. Therefore, individuals with strong GFP fluorescence were selected for passage. GFP fluorescence was observed throughout almost the whole worm body in the F1, F2, and later generations. Strong fluorescence was particularly observed in the pharynx (Figure 2b,c). All three strains displayed similar fluorescence patterns. During passage, some individuals showed a slightly bulging morphology (Figure 2d), infertility, or died after poor growth. These abnormalities roughly correlated with the intensity of GFP fluorescence.

The proteins extracted from the GFP-fluorescent *C. elegans* were separated with SDS-PAGE and subjected to Western blotting analysis with an anti-GFP antibody. A band of around 92 kDa was detected, approximating the size of the fusion protein combining PonAAS2 and GFP (Figure 3).

Next, we analyzed the levels of endogenous IAA in individuals expressing PonAAS2::GFP to confirm the expected functioning of PonAAS2. Because not all the progeny of the green fluorescent individuals displayed fluorescence, we picked three fluorescent individuals from each of the three PonAAS2::GFP lines in one tube and analyzed their IAA levels. This experiment was performed in triplicate, with wild-type *C. elegans* as the negative control. Whereas no IAA was detected in the wild type, IAA was detected in each PonAAS2::GFP line at 0.3–0.7 ng per individual (Figure 4a). When the activity converting Trp to IAA was measured in crude enzyme solutions prepared from these strains, the PonAAS2::GFP strains showed stronger conversion activities than the wild-type strain (Figure 4b), although the differences were not statistically significant. The lack of statistically significant differences may be attributable to the large sample-to-sample variability and the formation of IAA by non-enzymatic reactions (discussed further in the Discussion section). However, these results clearly indicate that PonAAS2 conferred IAA productivity on *C. elegans*.

### 3.2. Characterization of PonAAS2 Inhibitor

The initial screen identified several potential PonAAS2-inhibiting compounds (Appendix A). Appendix A only shows compounds with statistically significant inhibitory activities. Among these, HTS10550 (2-[3-(2,3-dihydro-1-benzofuran-5-yl)-1H-1,2,4-triazol-5-yl]phenol) and HTS09643 (2-[2-(3,4-dihydroxyphenyl)-2-oxoethyl]-1H-1,2-benzisothiazole-1,1,3(2H)-trione) appeared the most promising, with clear inhibitory activities for PonAAS2. The high production of IAAld in the presence of JA00082 suggested that the compound enhances the enzymatic activity of PonAAS2. However, further investigation revealed that the higher production of IAAld was actually due to a non-enzymatic reaction, confirmed by the production of IAAld simply by mixing and incubating the compound with Trp in the absence of PonAAS2. Although the mechanism was intriguing, it was not pursued further.

AASs and AAADs are enzymes that use aromatic amino acids as substrates and are highly homologous. Crystal structure analyses have provided information on the amino acid residues responsible for the differences between the AAAD and AAS activities [13,25] and on those involved in substrate recognition [25,26,27]. In the course of studying AAADs, researchers have also investigated the inhibitors of these enzymes, and it has been shown that AMD and carbidopa (Figure 5) are antagonistic inhibitors, which bind to AAADs competitively with their substrates [28]. Those compounds also display inhibitory activity against AAADs in insects [29]. Although AMD and carbidopa have not been studied as inhibitors of AAS, AMD is metabolized by an AAS of *D. melanogaster* [10]. This suggests that these compounds are candidate inhibitors of AAS. Because PonAAS2 recognizes Trp as a substrate, we examined the inhibitory activity of α-methyltryptophan (Figure 5), as well as AMD and carbidopa, against PonAAS2 and compared their inhibitory activities with those of HTS10550 and HTS09643 (Figure 5). As anticipated, the three known compounds showed significant inhibitory activities against PonAAS2, with AMD being the strongest. However, HTS10550 showed even stronger inhibitory activity than AMD. Therefore, we further characterized HTS10550 as an inhibitor of PonAAS2. The inhibitory activity of HTS10550 was investigated in detail at various concentrations, and the half maximal inhibitory concentration (IC_50_) was determined to be 1.0 μM (Figure 6a). A kinetic analysis indicated that HTS10550 exerts competitive inhibition, with a *K*i value of 0.20 μM (Appendix A).

The specificity of HTS10550 is crucial for its application in the inhibition of IAA biosynthesis in vivo. Therefore, we investigated whether HTS10550 also inhibits AAAD of the galling sawfly. In a previous study, we cloned PonAAS1 and PonAAS3, which shared high sequence homology with AAAD and AAS and showed that they did not convert Trp to IAAld [9]. AAS and AAAD function as dimers, and the catalytic residues are two amino acids located in the loops of the different monomers. In PonAAS2, these residues are asparagine 192 (N192) and tyrosine 332 (Y332). In plants, the amino acid residue corresponding to Y332 is crucial in distinguishing between AAS and AAAD functions. It has been shown that the *p*-hydroxy group on Tyr acts as a proton-donating group in enzymes that function as AAADs, whereas AASs usually have Phe at that position [25]. In plants, the amino acid corresponding to N192 in PonAAS2 is histidine (His) in both AAS and AAAD, with a few exceptions. However, in insects, the amino acid corresponding to Y332 is Tyr in all the AASs and AAADs that have been studied so far. The amino acid corresponding to N192 in PonAAS2 plays a key role in distinguishing between the functions of AAS and AAAD in insects [13]. Specifically, all insect AASs have asparagine (Asn) at this position, whereas all AAADs have His. Based on this information, it is highly likely that PonAAS1 and PonAAS3 are AAADs, because they both have His at position 192 and 193, respectively. In the present study, we re-examined the activities of PonAAS1 and PonAAS3 as AAADs and AASs using five aromatic amino acids (DOPA, Trp, Phe, Tyr, and 5-HTrp) as candidate substrates. While PonAAS1 showed clear activity in the conversion 5-HTrp→serotonin, neither PonAAS1 nor PonAAS3 showed AAS activity (Figure 7). PonAAS3 did not convert any of the five substrates to an amine or aldehyde. We then evaluated the inhibitory activity of HTS10550 against PonAAS1 and found that the IC_50_ value was about 20 μM (Figure 6b), indicating that HTS10550 inhibits PonAAS2 much more effectively than it inhibits PonAAS1.

Finally, we confirmed whether the inhibitor reduced IAA productivity in high-IAA-producing worms. There are six genes homologous to *AAS* in *C. elegans*, all of which were considered to be AAADs based on a phylogenetic analysis and the amino acids involved in the reactivity described above. Wild-type *C. elegans* showed no abnormal growth under normal growth conditions, even in the presence of 50 µM HTS10550. Therefore, PonAAS2::GFP-producing worms reared for more than two generations on solid medium in the presence or absence of HTS10550 were randomly collected and their endogenous IAA levels were determined (Figure 8). Their IAA levels were significantly reduced by the inhibitor. These results indicate that, at least in *C. elegans*, HTS10550 is efficiently incorporated into the worm and can inhibit PonAAS2 activity without the rapid inactivation of HTS10550.

## 4. Discussion

Although the involvement of phytohormones, such as auxin and cytokinin, in gall induction has been suggested, the evidence supporting this claim has been largely circumstantial, including the endogenous hormone levels in galling insects and the expression of marker genes in the gall tissues of host plants [1,2,4]. To provide experimental evidence of phytohormone involvement, it would be useful to either enhance phytohormone productivity in non-galling insects or disrupt phytohormone production in galling insects. In this study, we utilized *C. elegans* to demonstrate that the introduction of the *PonAAS2* gene resulted in high levels of IAA production in individual animals. Injecting a PonAAS2-encoding expression vector into *C. elegans* at high concentrations (e.g., 1000 ng/μL or 500 ng/μL) caused many individuals to die as juveniles with no growth, although they did emit GFP fluorescence. In contrast, when the plasmid was injected at 200 ng/μL, lines of *C. elegans* were established that survived for generations. However, even in these cases, we observed individual differences in the intensity of GFP fluorescence, and worms with strong fluorescence were often deformed or did not produce progeny, suggesting that IAAld produced in excess by PonAAS2 was toxic to the worms. This toxicity may be attributable to the high reactivity of the aldehyde group with the primary amino groups of important biomolecules, such as proteins (e.g., the free amino group on Lys). PonAAS2::GFP-expressing worms produced 0.3–0.7 ng of IAA per individual (Figure 4a). Based on a previous report, the average volume of IAA in adult individuals was estimated to be 4.5 nL [30]. Assuming a worm body density of 1 g/cm^3^, the concentration of IAA was calculated to be 70–160 μg/g fresh weight (fw), which is exceptionally high compared with the range of 300–1000 ng/g fw in galling sawfly larvae and 5–10 ng/g fw in the host plant leaf tissue [2]. Because insects typically display strong IAAld→IAA conversion activity [9], the introduction of PonAAS2 should provide insects with high IAA productivity. Although we conducted conversion experiments multiple times to compare the Trp→IAA conversion activity of PonAAS2::GFP-expressing worms with that of wild-type worms, we could not detect a statistically significant difference in conversion activity, despite consistently observing higher conversion in the PonAAS2::GFP-expressing worms (Figure 4b). The lack of a statistically significant difference can be attributed to the small amounts of [^13^C_10_,^15^N_1_]IAA produced in the wild-type worms and the large interindividual variation. Trp can be converted non-enzymatically to IAAld in the presence of FAD [8]. In our previous study of aldehyde oxidase, the enzyme that catalyzes the conversion of IAAld to IAA, we observed the conversion of Trp to IAAld, even with the heat-inactivated enzyme, whereas the IAAld→IAA conversion was completely abolished by heat treatment [7]. Similarly, a small amount of Trp→IAAld conversion is always observed in crude *E. coli* protein preparations, even after heat denaturation, although the mechanism of conversion is not known. Because we detected no endogenous IAA in wild-type worms but did observe weak Trp→IAA conversion, it is likely that such a non-enzymatic reaction also occurred in the extract of *C. elegans*.

The knockout of biosynthetic genes via genome editing or the suppression of their expression by RNA interference (RNAi) are possible strategies to disrupt phytohormone production in galling insects. However, these experimental systems have not yet been established in the galling sawfly. In this study, we suppressed IAA synthesis in vitro with HTS10550, identified as an inhibitor of PonAAS2 with relatively strong inhibitory activity. A kinetic analysis showed that HTS10550 inhibits PonAAS2 competitively. The dihydrobenzofuran ring of HST10550 may be recognized in place of the indole ring of the substrate Trp. HTS10550 inhibited PonAAS2 better than it inhibited PonAAS1, an AAAD of the sawfly, with a 20-fold lower IC_50_, so it may inhibit the biosynthesis of IAA without disrupting other essential physiological functions in insects, such as the biosynthesis of neurotransmitters. In fact, there were no growth problems in *C. elegans,* even in the presence of the inhibitor, and it seemed to have no effect on the AAAD family proteins in *C. elegans*. However, because the function of PonAAS3, which appears to be another AAAD, is not yet clear, we cannot exclude the possibility that HTS10550 inhibits the function of this enzyme.

In this study, we have shown that PonAAS2 can be used to initiate IAA production in IAA-non-producing *C. elegans*. This result demonstrated that PonAAS2 could confer high IAA productivity to low-IAA-producing organisms. Moreover, we developed an inhibitor of PonAAS2, thereby opening avenues for demonstrating the importance of IAA production in gall induction.

## Figures and Tables

**Figure 1 insects-14-00598-f001:**
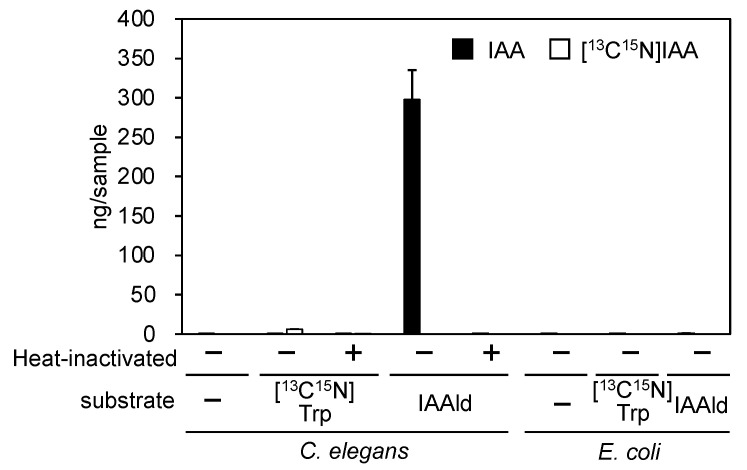
Endogenous IAA and Trp→IAA and IAAld→IAA conversion activities in *C. elegans* and *E. coli*, the food of *C. elegans* (mean ± SD).

**Figure 2 insects-14-00598-f002:**
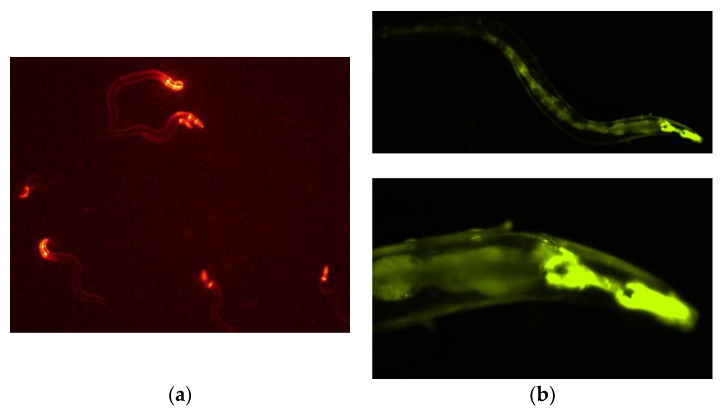
*Caenorhabditis elegans* expressing PonAAS2::GFP. (**a**) First screening of *C. elegans*, showing red fluorescence after injection with both expression vectors for the expression of PonAAS2::GFP and mCherry. (**b**) One of the lines stably displayed GFP green fluorescence for many generations. Lower panel, high-magnification view. (**c**) Bright-field image of the individual shown in (**b**). (**d**) Worm displaying abnormal morphology with strong green fluorescence.

**Figure 3 insects-14-00598-f003:**
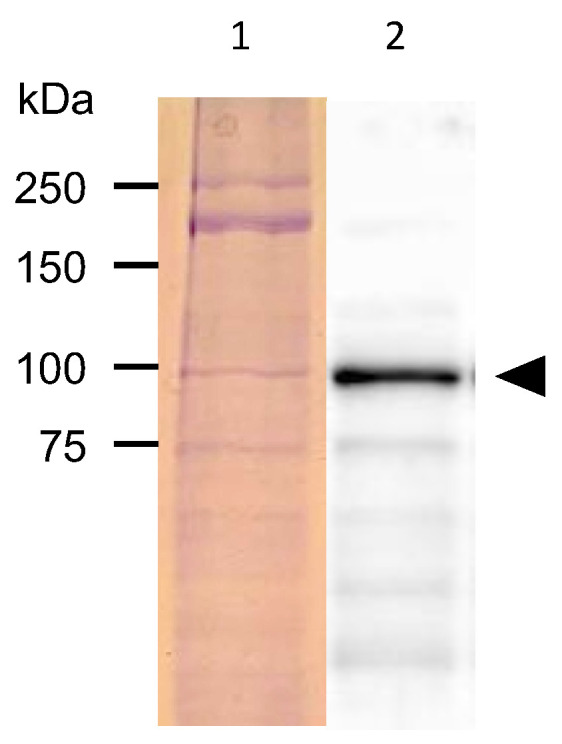
Detection of PonAAS2::GFP protein. Lane 1, proteins separated on SDS-PAGE and stained with Coomassie Brilliant Blue; lane 2, PonAAS2::GFP detected with anti-GFP antibody (arrowhead).

**Figure 4 insects-14-00598-f004:**
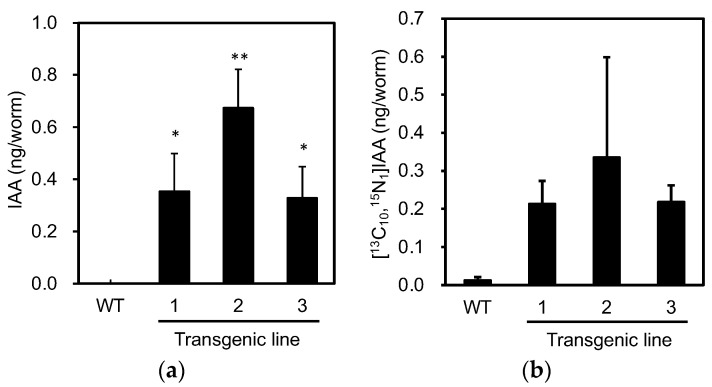
Characterization of IAA productivity in *C. elegans* expressing *PonAAS2::GFP*. (**a**) Endogenous levels of IAA in individual worms (mean ± SD). (**b**) Amounts of [^13^C_10_,^15^N_1_]IAA converted from [^13^C_11_,^15^N_2_]Trp in individual worms (mean ± SD). ** and * indicate significant differences at *p* < 0.01 and *p* < 0.05, respectively (Dunnett’s test, *n* = three biological replicates).

**Figure 5 insects-14-00598-f005:**
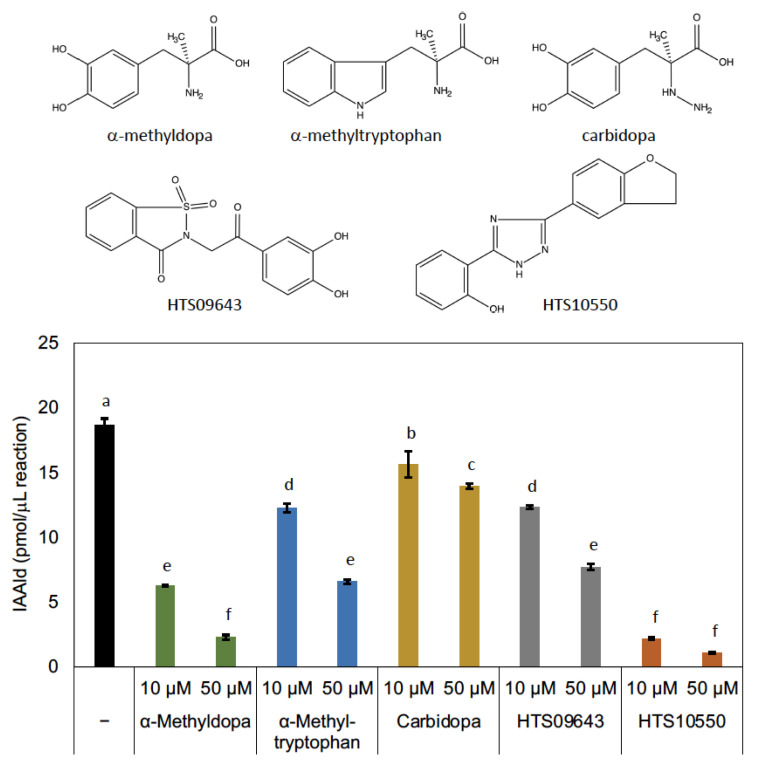
Structures of candidate inhibitors and their inhibitory activities against PonAAS2. Statistical significance was analyzed using ANOVA (*p* < 0.05) followed by Tukey’s multiple comparison test (*p* < 0.05). Different lower-case letters indicate statistically significant differences (*n* = three technical replicates).

**Figure 6 insects-14-00598-f006:**
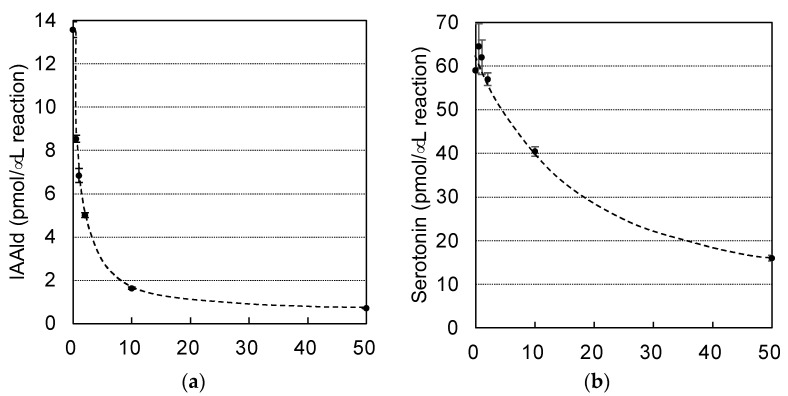
Comparison of the inhibitory activities of HTS10550 against AAS and AAAD of the galling sawfly. Dose–response inhibition curves for HTS10550 against the Trp→IAAld conversion by PonAAS2 (**a**) and the 5-HTrp→serotonin conversion by PonAAS1 (**b**).

**Figure 7 insects-14-00598-f007:**
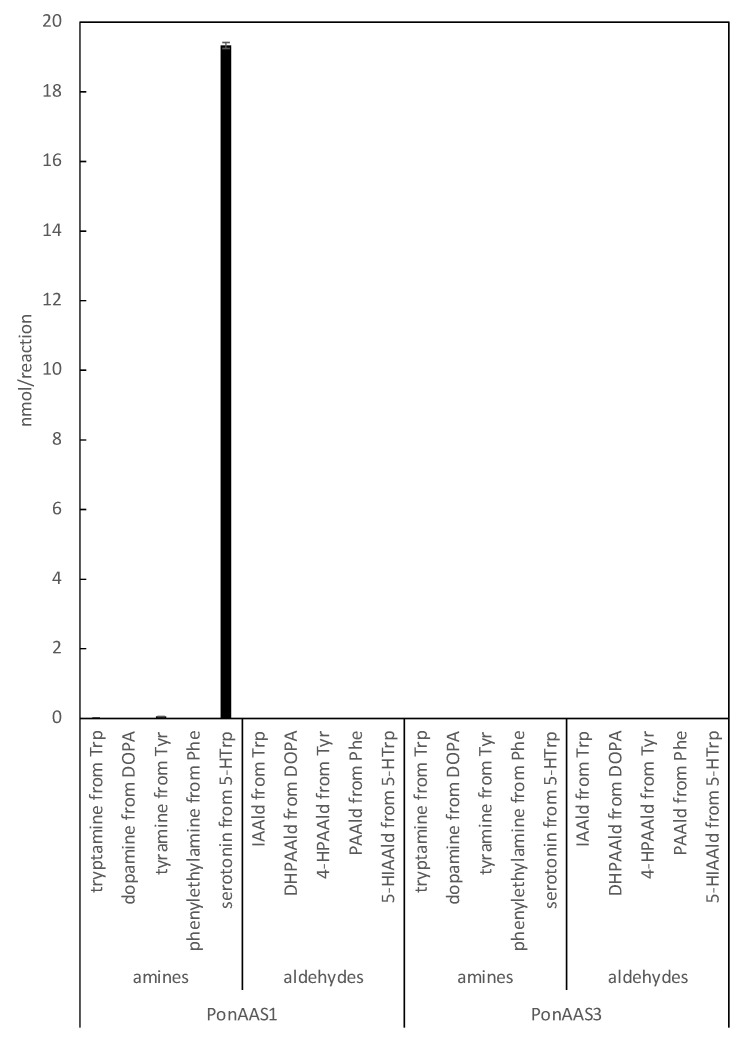
Enzyme activities of PonAAS1 and PonAAS3 as AAAD or AAS. Each of the five aromatic amino acids (Trp, DOPA, Tyr, Phe, or 5-HTrp) was incubated with the enzymes, and the corresponding amines (tryptamine, dopamine, tyramine, phenylethylamine, or serotonin, respectively) or aldehydes (IAAld, DHPAAld, 4-hydroxy PAAld [4-HPAAld], PAAld, or 5-hydroxy IAAld [5-HIAAld], respectively) were analyzed (*n* = three technical replicates).

**Figure 8 insects-14-00598-f008:**
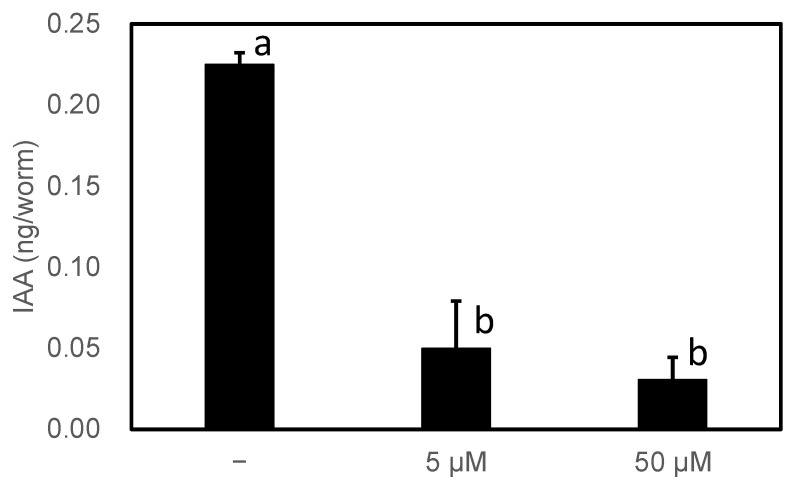
Effect of HTS10550 on endogenous IAA levels in high-IAA-producing *C. elegans*. Statistical significance was analyzed using ANOVA (*p* < 0.001), followed by Tukey’s multiple comparison test (*p* < 0.001). Different lower-case letters indicate statistically significant differences (*n* = three biological replicates).

## Data Availability

Sequences of PonAAS1, PonAAS2, and PonAAS3 have been deposited in the DNA Data Bank of Japan (PonAAS1, accession no. LC612383; PonAAS2, accession no. LC612384; PonAAS3, accession no. LC612385).

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
