# Peer review of "Conferring High IAA Productivity on Low-IAA-Producing Organisms with PonAAS2, an Aromatic Aldehyde Synthase of a Galling Sawfly, and Identification of Its Inhibitor"

_insects, 2023, doi:10.3390/insects14070598_

Round 1

Reviewer 1 Report

In this paper, the authors show that the introduction of the PonAAS2 gene from a gall-inducing sawfly can cause high IAA productivity in another organism.  They also identified and successfully tested a specific inhibitor of PonAAS2.  This paper provides noteworthy first steps in coming up with the methodology to definitively show whether IAA production by a gall-forming insect plays an important role in gall induction, and is of great significance in that regard. 

Line 20 of the Simple Summary  "and use it to demonstrate the importance of IAA in gall induction" should be removed as this was not accomplished.

My main suggestion for this paper is to remove some of the discussion that does not directly relate to what was accomplished in this study.  I strongly suggest that the text from line 432 to approximately 445 be deleted.  I would also delete approximately 449 to about 459.  Those sections of the discussion deal with experimental procedures that were not described in the Materials & Methods. While they address objectives of the researchers, they do not address the objectives for this specific paper. I think that those regions of the discussion detract from the paper and also raise some questions since the methods have not been fully described (that is why I indicated that the methods were not adequately described). I think it is sufficient to indicate what the ultimate goal is, and to mention that so far there have been some problems with applying these approaches to the sawfly gall system, but that the hope is to be able to do so in the future.   

Author Response

Dear reviewers: thank you for the valuable comments and suggestions. We have carefully examined all of the feedback, and have revised the manuscript accordingly.

[comment]

In this paper, the authors show that the introduction of the PonAAS2 gene from a gall-inducing sawfly can cause high IAA productivity in another organism.  They also identified and successfully tested a specific inhibitor of PonAAS2.  This paper provides noteworthy first steps in coming up with the methodology to definitively show whether IAA production by a gall-forming insect plays an important role in gall induction, and is of great significance in that regard. 

Line 20 of the Simple Summary "and use it to demonstrate the importance of IAA in gall induction" should be removed as this was not accomplished.

[response]

As per the reviewer's suggestion, we have eliminated the relevant part from the article.

[comment]

My main suggestion for this paper is to remove some of the discussion that does not directly relate to what was accomplished in this study.  I strongly suggest that the text from line 432 to approximately 445 be deleted.  I would also delete approximately 449 to about 459.  Those sections of the discussion deal with experimental procedures that were not described in the Materials & Methods. While they address objectives of the researchers, they do not address the objectives for this specific paper. I think that those regions of the discussion detract from the paper and also raise some questions since the methods have not been fully described (that is why I indicated that the methods were not adequately described). I think it is sufficient to indicate what the ultimate goal is, and to mention that so far there have been some problems with applying these approaches to the sawfly gall system, but that the hope is to be able to do so in the future. 

[response]

In response to the reviewer's feedback, we have revised the section on future objectives and issues to provide clearer and more concise descriptions. We have taken the suggestion to simplify the content, as I acknowledge that I had initially included excessive details. (L433-439)

Reviewer 2 Report

This study provides valuable findings about the how phytohormones like IAA play an important role in gall induction by gall inducing insects. The use of C. elegans as model organism to see how IAA is produced in these non-IAA producing organism by injecting PonAAS2 (a gene responsible for IAA production in sawflies) show how the role of this gene in galling insects. Also use of inhibitor for IAA showed how this phytohormone interacts.

The manuscript is well prepared and presents the finding in a clear and concise manner. However, few minor suggestions can improve the manuscript

1. Line 193: How were these E.coli cells taken from the worms, do explain briefly about it

2. Line 186, 231...: Use of 'several' and 'plenty' is very arbitrary. Do mention the numbers or the counts of the worms for a better understanding.

Author Response

Dear reviewers: thank you for the valuable comments and suggestions. We have carefully examined all of the feedback, and have revised the manuscript accordingly.

[comment]

This study provides valuable findings about the how phytohormones like IAA play an important role in gall induction by gall inducing insects. The use of C. elegans as model organism to see how IAA is produced in these non-IAA producing organism by injecting PonAAS2 (a gene responsible for IAA production in sawflies) show how the role of this gene in galling insects. Also use of inhibitor for IAA showed how this phytohormone interacts.

The manuscript is well prepared and presents the finding in a clear and concise manner. However, few minor suggestions can improve the manuscript

  1. Line 193: How were these E. coli cells taken from the worms, do explain briefly about it

[response]

The E. coli was not taken from C. elegans, but was cultured as food for the nematode. I think this part was misleading, so I have added a phrase that makes it clear that the E. coli serves as the food for the nematode. (L195)

[comment]

  1. Line 186, 231...: Use of 'several' and 'plenty' is very arbitrary. Do mention the numbers or the counts of the worms for a better understanding.

[response]

L187: The exact count of worms was not recorded, but it was estimated to be around ten. Therefore, in the revised manuscript, I described the number as "about ten."

L233-234: We apologize for the lack of information regarding the exact number of worms used in this study. Based on observations, it is estimated to exceed 2,000 worms. As indicated in the experimental section, this quantity corresponds to approximately 0.2 of a plate saturated with worms after one week of culture. Therefore, we have represented the worm quantity in this manner.